# Biotechnological Utilization of Agro-Industrial Residues and By-Products—Sustainable Production of Biosurfactants

**DOI:** 10.3390/foods13050711

**Published:** 2024-02-26

**Authors:** Damjan Vučurović, Bojana Bajić, Zorana Trivunović, Jelena Dodić, Marko Zeljko, Rada Jevtić-Mučibabić, Siniša Dodić

**Affiliations:** 1Department of Biotechnology, Faculty of Technology Novi Sad, University of Novi Sad, Bulevar cara Lazara 1, 21000 Novi Sad, Serbia; dvdamjan@uns.ac.rs (D.V.); ron@uns.ac.rs (Z.T.); klik@uns.ac.rs (J.D.); marko.zeljko@uns.ac.rs (M.Z.); dod@uns.ac.rs (S.D.); 2Institute for Food Technology Novi Sad, University of Novi Sad, Bulevar cara Lazara 1, 21000 Novi Sad, Serbia; rada.jevtic@fins.uns.ac.rs

**Keywords:** biosurfactants, biotechnology, agro-industrial residues, agro-industrial by-products, biodegradable waste

## Abstract

The importance and interest in the efficient use and valorization of agro-industrial residues and by-products have grown due to environmental problems associated with improper disposal. Biotechnological production processes, including microbial biosurfactant production, represent a sustainable way to utilize agro-industrial residues and by-products, which are applied as substrates in these processes. Biosurfactants produced by microorganisms using renewable resources are a viable alternative to traditional petrochemical surfactants and have several potential uses in a wide range of industrial sectors due to their minimal ecotoxicity, easy biodegradability, and moderate production conditions. The common applications of biosurfactants, besides in food industry as food additives and preservatives, are in agriculture, environmental protection, the cosmetics and pharmaceutical industry, wastewater treatment, the petroleum industry, etc. This review aims to summarize the comprehensive scientific research related to the use of various agro-industrial residues and by-products in the microbial production of biosurfactants, as well as to emphasize the present state and the importance of their sustainable production. Additionally, based on the available biosurfactant market analysis datasets and research studies, the current situation in science and industry and the future perspectives of microbial biosurfactant production have been discussed.

## 1. Introduction

In recent years, due to environmental problems, the importance and interest in the efficient use and valorization of agro-industrial residues and wastes has grown [1]. Agriculture-related industries produce an immense quantity of effluents each year that, if released into the environment without proper treatment, could negatively impact both human and animal health, as well as the environment [2]. Regardless of their biodegradability, the vast quantities in which they are generated require finding adequate solutions for their utilization [3]. Agri-food wastes possess a complex structure and composition comprising polysaccharides, proteins, carbohydrates, polyphenolic compounds, and other components. These wastes are cost-effective, readily available, environmentally friendly, and renewable natural resources [4]. There are a large number of biotechnological solutions that can be applied to solve the problem of waste. In addition to being economically profitable, they can easily be applied while positively affecting the protection of the environment and human health [5].

Regarding the potential development of agro-industrial residue, waste, by-product, or wastewater valorization processes, it is important to systematically evaluate the production process and characteristics of agro-industrial wastes/residues/by-products, choose one or more goals (e.g., recycling/recovering components), examine the economic and technology benefits, choose the potential market, and identify the bioprocess solution to accomplish the desired aim [6]. Food industries generate a large amount of agri-food waste at the level of agricultural crops, as well as during processing, with around 1.6 billion tons of primary product equivalents, and the total food waste for the edible part of this amounts to 1.3 billion tons. When produced on an industrial scale, the benefit of utilizing waste or by-products from the agri-food industry as raw materials is a reduction in manufacturing costs, due to the fact that the waste does not have a potential use [7]. Using agro-industrial residues and by-products as raw materials for producing products with an enhanced value has created novel opportunities that contribute to the preservation of the environment. Incorporating agro-industrial residues and by-products as a substrate substantially reduces the overall process expenditure, making the production more environmentally friendly and in line with the principle of a circular bioeconomy. This scenario favors any bioprocess that utilizes residues or by-products from the agriculture and food industry while simultaneously reducing production costs and waste generation [8,9,10].

In recent decades, scientists around the world have been working to find solutions to reduce generated agro-industrial residues and wastes through the improvement and development of the biotechnological production of various high-value products. Different agro-industrial residues, wastes, and by-products are used for the production of different bioproducts. For example, soybean molasses has been used for the production of bioethanol [11], biogas [12], and extracellular polymers [13]; sugar beet molasses for single cell proteins [14], bioethanol [15,16], and biohydrogen [17]; sugar beet pulp for single cell proteins [14], and bioethanol, biomethane, and biohydrogen [18]; wheat chaff for cellulase and xylanase [19], and itaconic acid [20]; corn straw for bioethanol and biogas [21], fumaric acid [22], and lactic acid [23]; and wheat straw for lactic acid [24], bioethanol, biohydrogen, and biogas [25]; while wastewaters obtained in the different industrial processes were used for the production of lipases, proteases, and tannases [26], xanthan [27,28], bioethanol [29], and biomass [30], etc.

In addition to the above, various agro-industrial residues and by-products can be used in the production of biosurfactants. Biosurfactants, created by micro-organisms using renewable resources, are a viable alternative to traditional petrochemical surfactants [10,31,32,33,34]. In their review, Mohanty et al. [32], as adequate substrates for the production of biosurfactants, described agro-industrial waste and its by-products, including fruit and vegetable waste, and starch-rich waste, industrial waste, lignocellulosic waste, oily and glycerol-based waste, and other substrates, including frying oil waste, waste from vegetable oil processing and its by-products, dairy industrial waste, and sugar industrial waste. According to Santos et al. [33], several frequently utilized industrial wastes as raw material for the biosynthesis of biosurfactants include olive oil mill effluent, animal fat, frying oils, oil cakes, sugarcane and beet molasses, whey, corn steep liquor, and starchy substrates.

This review aims to present the current situation and future perspectives of biosurfactant production and to consolidate the data published in the scientific literature in recent years related to the use of various agro-industrial residues and by-products in the production of biosurfactants, and the most significant characteristics of their production and application.

## 2. Surfactants vs. Biosurfactants

### 2.1. Adventages and Disadventages

Figure 1 summarizes all of the succeeding comparisons made between biosurfactants and surfactants, besides being a schematic presentation of a micelle cluster of surfactants.

Both surfactant and biosurfactants are compounds that possess a hydrophilic polar head, interacting with water, and a hydrophobic nonpolar tail, refraining from interactions with water, in their structure, or as what are commonly referred to as amphipathic molecules [35]. They are categorized into four categories based on the charge of their hydrophilic group: anionic (negatively charged), cationic (positively charged), nonionic (without any charge), and ampholytic/zwitterionic (both charges) [36]. The hydrophobic part of surfactants is normally a hydrocarbon (linear or branched), whereas the hydrophilic part consists of ionic or strongly polar groups, such as sulfate; sulfonate; phosphate; carboxylate salts; protonated primary, secondary, and tertiary amines; quaternary ammonium salts; and polyethylene oxide [37]. Biosurfactants, however, exhibit a broader range of chemical compositions; i.e., they have better structural diversity when compared to synthetic surfactants [38]. The polar hydrophilic portion (head group) can consist of sugars, amino acids, peptides, proteins, or polar functional groups such as carboxylic acid, while the hydrophobic component may consist of saturated, unsaturated, and/or hydroxylated fatty acids or fatty alcohols [39].

This specific amphipathic structure makes them possible to adsorb at the liquid interface, thus forming a monolayer, which lowers the cohesive forces between liquid’s molecules (that act as a thin, elastic membrane), and, as a result, the surface tension is lowered. In this way, surfactants have the ability to reduce the surface tension between two materials, such as a gas and a liquid (water surface tension), two liquids (water and oil), or a liquid and solid (water and dirt particles). The origin of the word surfactants describes just that: surfactants is a contraction of SURFace-ACTive AgeNTS, hence their use in these types of two-phase systems. Again, due to their structural, as well as functional, variety, biosurfactants have found a wider utilization spectrum. Biosurfactants have several potential applications in a wide range of industrial sectors such as in the food industry as food additives and preservatives, and are used as emulsifiers, as thickeners and stabilizers, and a wetting agent, for antioxidant and antimicrobial applications [40,41]; in the petroleum industry for the extraction of crude oil from reservoirs, the formulation of fuels, biodesulphurization, the transport of crude oil by pipelines, and oil storage tank cleaning [42]; in environmental protection for bioremediation, oil spill cleanup operations, and soil remediation and flushing [33,43]; in oil–water separation, contaminant degradation, effluent flotation, and heavy metal remediation [40]; in agriculture for the elimination of phytopathogens, agricultural soil remediation, crop protection, and pesticide formulations [44]; in the cosmetics and pharmaceutical industry for detergents, emulsifiers, stabilizers, biofilm controlling, antibacterial applications, and antioxidant, moisturizing, healing, and skin-toning properties [40,41]; and in nanotechnology for stabilization and reducing nanoparticle formulation [45].

The critical micelle concentration (CMC) refers to the minimum concentration of a surfactant in a bulk phase at which micelles, which are aggregation of surfactant molecules, begin to form. In these micelles, the hydrophilic heads of the surfactant molecules are distributed along the perimeter of the sphere, while the hydrophobic tails are pointed toward its core (schematic presentation in Figure 1). In many processes, the CMC specifies the limiting concentration for meaningful use. When the formation of micelles is desirable, e.g., when cleaning, the CMC is a measure of the efficiency of a surfactant. Surfactants with lower CMC values are more effective. Biosurfactants have 40% lower CMC than synthetic surfactants [38]. The ratio of the hydrophilic portion of the molecule to the hydrophobic portion, known as the hydrophilic–lipophilic balance (HLB), affects the efficacy of surfactants’ wetting, anti-foaming, and emulsifier properties [46].

The broader compositional diversity of biosurfactants over surfactants can be attributed to the raw material they are being produced from, leading to another classification, which divides synthetic surfactants from natural or “green” surfactants (biosurfactants). The majority of synthetic surfactants are obtained in the petrochemical sector from non-renewable sources [41]. On the other hand, biosurfactants can be obtained from plants and animals in chemical (extractions, precipitation, or distillation) or enzymatic processes, as well as metabolism products of bacteria, yeast, and fungi when cultivated (fermented) on different renewable resources [47]. In terms of the United Nations sustainable development goals in building a green economy, biosurfactants are obviously favored as products generated in a sustainable fashion due to utilizing renewables.

### 2.2. Classification of Biosurfactants

Besides their diverse chemical composition, biosurfactants can be produced by a wide variety of distinct microorganisms. These metabolites are produced as secondary (in the stationary growth phase) extracellular compounds roaming freely in the cultivation medium or remain attached to the microbial cell surface [38]. Additionally, the producing microorganisms have the ability to synthesize a wide range of biosurfactants concerning their molecular weight, which can vary from low to high. The broad variety of biosurfactants regarding their microbial origin, chemical structure, and molecular weight is presented in Figure 2.

As can be seen from Figure 2, some producing strains (underlined) have the ability to produced biosurfactants with a different chemical composition and molecular weight, depending on the carbon source present in the cultivation medium. Low-molecular-weight surfactants, which include glycolipids (such as rhamnolipids, sophorolipids, and mannosylerythritol lipids), lipopeptides (such as surfactin and lichenysin), and phospholipids, and high-molecular-weight surfactants, which include polymeric (such as liposan, rufisan, and emulsan) and particulate surfactants, are the two primary classes of biosurfactants [48]. However, Sarubbo et al. [41] classified the high-molecular-weight biosurfactants as bioemulsifiers due to their capability to efficiently stabilize oil–water emulsions.

Microbial biosurfactants are produced at milder operating conditions (pH, temperature, salt concentration, etc.) compared to synthetic surfactants [49]. Likewise, biosurfactants are highly superficial molecules that can work in a broad range of pH (2.0–12.0), temperature (30–60 °C), and salinity (to 20% NaCl) conditions [32]. An additional advantage of biosurfactants compared to synthetic surfactants is their low toxicity, biodegradability, and eco-friendliness [36,50]. This property is attributed to their structural components; i.e., they are made of biomolecules, do not accumulate in the environment, and are easily degraded by natural processes [51]. Sarubbo et al. [41] reported that replacing synthetic surfactants with biosurfactants would reduce lifetime CO_2_ emission by 8%. Similarly, Foley et al. [46] estimated that, if renewable surfactants were to entirely replace petrochemical surfactants in the EU, the total CO_2_ emissions associated with surfactant production and use could be reduced by as much as 37%. Since biosurfactants occupy only up to 10% of the total world production of surfactants [41], there is a long way to go. The antibacterial, antifungal, and antiviral properties of biosurfactants in combination with an antiadhesive effect provide biocompatibility. This property makes biosurfactants suitable for extensive applications in pharmaceuticals and personal care products, and as food additives [52]. Due to the presence of unique functional groups like sugar or peptide moiety in their hydrophilic parts, surfactants could be a good candidate for specificity-related applications such as drug delivery, the emulsification and detoxification of contaminants in industrial waste, and gene transfection [32].

Besides all of the aforementioned advantages, the manufacturing cost of biosurfactants is 3 to 10 times costlier than the production of their chemical counterparts [53]. Several authors have suggested that concentrating on four significant factors could reduce production costs, including the process feedstock, the surfactant-producing microorganisms, the production process, and the by-products [51,53]. Thus, the establishment of methodologies that facilitate the manufacturing and subsequent utilization of biosurfactants on a large scale is of paramount significance.

## 3. Agro-Industrial Residues and By-Products for Biosurfactant Production

### 3.1. Clasification of Agro-Industrial Residues and By-Products

All residues and by-products from forestry, farm animals, agriculture, and industries that are based on agriculture are referred to as agro-industrial wastes [5]. In the analysis of Zihare et al. [54], agro-industrial by-products are classified, based on the metabolic capability of microorganisms, into agricultural residues and industrial residues. Agricultural residues can be divided into sources rich in protein or lipids, starch, structural polysaccharides, and mono- and disaccharides, while industrial residues can be categorized as sources of polymers, carbon compounds, and sources of microorganisms that are photosynthetic. According to Sadh et al. [55], agro-industrial wastes can be divided into agricultural residues, which can be further divided into field residues (such as stalks, leaves, stems, etc.) and process residues (such as husks, seeds, molasses, roots, etc.), and industrial residues (such as potato peel, orange peel, cassava peel, coconut oil cake, soybean oil cake, etc.). Raut et al. [56] reported that there are four types of agricultural waste: crop (such as bagasse, husks, and straws), processing (such as fertilizer containers and materials for packaging), hazardous (such as chemicals for pest control) and animal (such as excrements and carcasses) waste, obtained after cultivating and processing fruits, dairy, grains, meat, and other agricultural products. Cuadrado-Osorio et al. [57] classified the agro-industrial residues based on their chemical composition into the following: the lignin and non-starchy structural polysaccharides group that is further divided into lignin-rich (such as cocoa bean shell and oil palm pit), hemicellulose-rich (such as banana pseudo stem and leaves, and corn stalks), and cellulose-rich (such as cocoa pod husk, coffee pulp, and corn cob) residues, the starchy group (such as cassava bagasse and stem, rice bran, and corn bran), and monosaccharides and free polysaccharides group (such as cocoa sweatings and molasses).

Taking into account the short literature overview given above, it is evident that there is no unique classification of bio-based resources derived from agriculture and industry. In this review, residues and by-products generated by different agricultural and industrial activities have been divided into four main types based on their origin: agricultural field residues (such as corn stalks, sunflower stalks, wheat straw, rice straw, oat straw, etc., remaining in the field after crop harvesting), agricultural process residues (such as corncobs, wheat chaff, rye chaff, rice chaff, coffee husk, groundnut husk, etc., obtained after crop processing), industrial residues (potato peels, apple peels, tomato pomace, grape pomace, frying oil, wastewaters, etc., generated by the food industry after the primary processing of raw material), and industrial by-products (molasses, whey, crude glycerol, bagasse, oil cakes, etc., produced by the food industry at the end of processing).

### 3.2. Agro-Industrial Residue and By-Product Selection Criteria

Due to their biodegradable character and composition, agro-industrial residues and by-products, such as those obtained from fruit, dairy, sugar, and oil processing industries, can be adequate raw materials for biosurfactant production if they fulfil all mandatory and most supplementary requirements. Considering that, as in every biotechnological production process, the producing microorganism requires balanced cultivation media in order to grow and produce the desired product, the ideal raw material for biosurfactant synthesis should have a high content of carbohydrates and/or lipids that are in a suitable form. Further, the selection of an appropriate raw material depends on its availability and price, which significantly affects the cost of the entire production process, and, therefore, the price of the final product [34]. Approximately 30–40% of the overall expenses are attributed only to the formulation of media for microorganism growth and biosurfactant production [58,59]. Consequently, alongside the proper content of a suitable carbon source(s), a raw material that is adequate for biosurfactant production should be renewable, cheap, readily available, easy to store, and accessible to combine with other residues and/or by-products [34]. Moreover, the alternative raw material should have a uniform quality, which facilitates the standardization of medium preparation and product separation, and it is desirable that the quantity of residues and by-products generated after its exploitation be minimal. Moreover, the usage of raw materials that contain thermolabile components should not be practiced because sterilization is unavoidable step in the medium preparation procedure [60].

In order to select the appropriate raw material for biotechnological production at an industrial scale, from the vast number of agro-industrial residues and by-products, first of all, it is necessary to identify the main raw material components and to determine their concentration, and, then, to comprehensively consider the biological availability of the identified components (suitability for application in producing microorganisms), raw material consistency, prevalence, seasonal availability, alternative applications, and local technological potential [61]. Table 1 shows the criteria that must be considered when selecting suitable agro-industrial residues and by-products as raw materials in the biotechnological production of biosurfactants, whereby the analysis of some typical representatives of each group of alternative substrates is given. Of those mentioned above, only the general criteria that come from the characteristics of the appropriate agro-industrial residues and by-products were considered. In contrast, the local technological potential and the location of raw material generation depend on factors that are beyond the scope of this study. However, it is important to note that the generation location is a significant criterion since each geographic area has specific raw materials. Therefore, based on the location of the biosurfactant production plant, the selection of those raw materials that are positioned at an acceptable distance will be more likely.

Raw materials, which are used as alternative substrates in biotechnological production, are sources of several nutrients, but, in practice, those that are primarily sources of carbon are the most often exploited. Information for the selected raw materials utilized in biosurfactant production as carbon sources are summarized in Table 1. Besides them, distillation stillage and corn steep liquor are interesting substrates for the discussed bioprocess. However, these raw materials are low-carbon sources, and may be added to the medium for biosurfactant production as nitrogen sources to substitute the costly yeast extract and other nitrogen compounds [59,62].

The prime criterion for the selection of a raw material with potential valorization in biosurfactant production is the biological availability of the detected carbon sources, i.e., their suitability for application in producing microorganisms, that can be high, moderate, or low. According to this criterion, it is most preferable that carbon compounds presented in specified raw materials be directly consumed by all biosurfactant-producing microorganisms (high biological availability), or at least by certain strains (moderate biological availability). It is also appropriate if carbon compounds can be easily converted into consumable substances using techniques that are cost-effective and ecologically justified [61]. Nevertheless, many agro-industrial residues and by-products contain complex carbon sources (mainly cellulose, hemicellulose, and lignin) whose biological availability is low, and the pretreatment is often complicated [32]. Nevertheless, the application of lignocellulosic substrates in biosurfactant production is wide due to their low market value [10]. The consistency of the raw material, which can be solid, oily, syrupy (concentrated liquid), or a diluted liquid, depending on the carbon source concentration, is another important selection criterion because its knowledge can facilitate manipulation during transport and/or storage and minimize composition changes and spoilage. The seasonal availability is usually considered together with the consistency. Raw materials can be available throughout the year or only during a certain season/campaign, and, based on the consistency, its seasonal availability can be very short (for wet solids and diluted liquids), if inexpensive and simple conservation is not provided, or prolonged (for dry solids and oily and syrupy liquids). However, regardless of the consistency and seasonality, it is important to note that the quantity and composition of the selected agro-industrial residues and by-products are not strictly uniform, which affects the continuous availability of raw materials of uniform quality. This must be taken into account when defining the procedure for the biosurfactant production medium preparation. The alternative application of selected raw materials is acceptable only if it will not significantly affect their competitiveness and market rate [60,61].

Table 2 shows the literature review of the utilization of various agro-industrial residues or by-products that are used as sole or partial substrates in the production of different classes/types of biosurfactants.

From analyzing Table 2, it can be concluded that, from the broad variety of raw materials and their combinations which can be used for cultivation media formulation, as well as numerous producing microbial strains which can be applied, a wide spectrum of biosurfactants with different structural and functional properties can be produced.

## 4. Sustainable Production of Biosurfactants from Agro-Industrial Residues and By-Products

The sustainable production of biosurfactants, using agro-industrial residues and by-products, has many benefits; i.e., it is cost-effective, non-toxic, and environmentally friendly, while simultaneously struggling with several challenges that include the availability and pretreatment of the raw material, downstream processing, and large-scale production [10,32]. The biotechnological process for the production of biosurfactants, as well as many other bioproducts, consists of several phases, which include the preparation for the bioprocess (the selection of the production microorganism and preparation of the inoculum, preparation of the media, sterilization of the media and equipment, etc.), performing the bioprocess in a suitable bioreactor under optimal process parameters, and the procedures for separation and purification in order to obtain the desired product. A schematic representation of the structure of a sustainable bioprocess for the production of biosurfactants from agro-industrial residues and by-products is given in Figure 3.

Parameters that significantly affect the growth of the production of microorganisms and the production of biosurfactants are agitation, aeration, temperature, fermentation time, pH value, and adequate foam control. Additionally, the selection of the appropriate production of microorganisms that gives a high yield of the desired biosurfactant, a suitable size of the inoculum, and the selection of an adequate and cheap substrate, i.e., the source of carbon and nitrogen and their appropriate ratio, are of great importance [82].

Aerobically grown microorganisms, such as yeasts, bacteria, and filamentous fungi, are usually used for the production of biosurfactants [83], where the most frequently studied microorganisms are from genera *Pseudomonas*, *Bacillus*, *Candida*, *Rhodococcus*, and *Corynebacterium* [84]. Bacteria are excellent producers of biosurfactants; however, due to their potential pathogenicity, those biosurfactants are unsuitable for application in the food industry. On the other hand, yeasts like *Saccharomyces cerevisiae*, *Kluyveromyces lactis*, and *Yarrowia lipolytica* have GRAS (Generally Recognized As Safe) status and pose no toxicity or pathogenicity hazards [33]. Figure 2 and Table 2 contain bacteria, yeasts, and filamentous fungi that are used in the production of biosurfactants.

The macronutrients, micronutrients, and trace elements are necessary for the synthesis of biosurfactants and represent a critical constituent in the formulation of an appropriate biosurfactant production medium [59]. Carbon, nitrogen, phosphorus, and mineral sources can affect the quality, quantity, and type of biosurfactants produced [85]. The vast majority of biosurfactants are produced in aqueous media utilizing suitable raw materials, such as oils, fats, carbohydrates, hydrocarbons, or combinations of these [83]. Biosurfactant composition is influenced by the carbon and nitrogen source [77]. Water-soluble raw materials (carbohydrates) tend to be utilized by the microorganism to create a hydrophilic moiety of the biosurfactant, whereas hydrophobic raw materials (fats and oils) are used to create the hydrophobic part of the biosurfactant [59]. In their research, Araújo et al. [77] used *S. marcescens* for the production of the biosurfactant, and lactose and corn oil were used as the carbon source, while cassava flour wastewater was used as the nitrogen source. In the cultivation medium, the C/N ratio should be high (10–40) because biosurfactants are usually secondary metabolites [86]. The addition of metal salts significantly affects the production of biosurfactants [86] and Gudiña et al. [80] showed, that when the culture medium was supplemented with the optimal concentrations of iron, manganese, and magnesium, biosurfactant production increased. Table 2 shows different agro-industrial residues and by-products used as substrates for the production of biosurfactants.

The availability and pretreatment of the raw materials, coupled with transportation costs, are the main issues with alternate raw materials, such as agro-industrial residues and by-products. Pre-treatment techniques applied for these raw materials for biosurfactant production can be divided into physical (mechanical extrusion, milling, freezing, ultrasound, pyrolysis, and microwave), physico-chemical (steam, hot water, wet oxidation, and ammonia-based), and chemical (acid, alkali, ozonolysis, ionic liquids, and solvents) [34]. The utilization of pretreatment procedures (size reduction, pre-hydrolysis, hydrolysis, and drying) can vary depending on the specific raw material, and a series of pretreatment is recommended for a wide variety of substrates, varying from oil-based substrates, starch substrates, lignocellulosic substrates, etc. [32].

According to Kanwal et al. [85], the bioreactor configuration and mode of operation are among the factors that are significant for the yield, productivity, and quality of biosurfactant production, and the bioreactor types that are used as mass production systems are continuous stirred tank bioreactors, pneumatically mixed bubble column bioreactors and airlift bioreactors with an internal or external loop, fluidized-bed bioreactors, packed-bed bioreactors, and photobioreactors. Biosurfactants can be produced by the well-known and widely used submerged or liquid fermentation (SmF) and solid-state fermentation (SSF), that is increasingly used [87]. According to Eras-Muñoz et al. [84]. the majority of the existing literature focuses on the submerged production of bacterial biosurfactants at the laboratory scale, employing small volumes in stirred Erlenmeyer flasks. However, there are also some studies that have explored the use of bioreactors and solid-state cultivation techniques. The SSF technique was utilized by Zhu et al. [88] for the production of lipopeptides by *Bacillus amyloliquefaciens* using rice straw as a major substrate, while Jiménez-Peñalver et al. [79] used the winterization oil cake and molasses for the production of sophorolipids with *Starmerella bombicola*.

Biosurfactant application is affected by the selection of the separation and purification procedures, and one way to achieve the efficient downstream processing for biosurfactants is to combine various approaches and unit operations to create a comprehensive strategy that would produce the purified desired bioproduct. According to Venkataraman et al. [89], the separation of biosurfactants consists of centrifugation (in order to remove the biosurfactant from the biomass supernatant); the precipitation and phase separation process (for partial purification); solvent extraction, ultrafiltration, and crystallization (for purification); and several different types of chromatographic techniques (for ultra-purification).

## 5. Current State and Future Perspectives of Biosurfactant Production

### 5.1. Biosurfactant Market

Biosurfactants have attracted the interest of the many end-use industries, resulting in an increasing demand for these bioproducts and a constant market expansion [90,91]. Numerous market research companies with global coverage are focused on biosurfactant market analysis, and the most important data from reports published by the leading institutions in this field are summarized in Table 3.

It is obvious that the values of the biosurfactant market size for the previous year differ to a large extent and that the projected values do not depend only on the defined forecast periods. This heterogeneous information is a result of the applied methodology for the collection and analysis of data, conducted individually by each company within the specified market and geography, which means that the same countries and/or biosurfactant-producing companies may not be covered. Nevertheless, it is evident that the global biosurfactant market accounted for more than USD 8 billion in 2022 and is expected to grow in the next 5–10 years at an annual rate of a minimum of 3.9%, to account for more than USD 14 billion in 2032. However, the biosurfactant market is still significantly lower than the chemical surfactant market. Fact.MR [102] reported that microbial biosurfactants hold only around 2.8% of the global surfactant market.

The global biosurfactant market comprises small- and medium-sized enterprises, as well as large companies. Many of them follow the latest market trends and invest in research and development to provide high-quality products and fulfil customer demand. In addition, players operating in the biosurfactant market focus on the adoption of next-generation technologies, creation of new products, and expansion of capacity, as well as establishing a partnership and collaboration in order to stay competitive in the market [91,100]. The important industrial biosurfactants producers are Evonik Industries AG (Germany), Henkel AG & Co. KGaA (Germany), Biotensidon GmbH (Switzerland), Givaudan International S.A. (Switzerland), Clariant (Switzerland), Ecover (Belgium), Solvay S.A. (Belgium), WHEATOLEO (France), AkzoNobel N.V. (Netherlands), Sabo S.p.A. (Italy), Boruta-Zachem SA Kolor (Poland), Holiferm, Ltd. (UK), Croda International Plc (UK), TeeGene Biotech, Ltd. (UK), Dow, Inc. (USA), BASF SE (USA), Jeneil Biotech, Inc. (USA), Stepan Company (USA) AGAE Technologies, LLC (USA), GlycoSurf (USA), Kemin Industries, Inc. (USA), TensioGreen (USA), Biosurfactants, LLC (USA), Dispersa, Inc. (USA) Synthezyme, LLC (USA), DuPont (USA), Ashland (USA), Locus Fermentation Solutions (USA), Innospec (USA), Deguan Biosurfactant Supplier (China), Daqing VICTEX Industries Co., Ltd. (China), Urumqi Unite Bio-Technology Co., Ltd. (China), Allied Carbon Solutions Co., Ltd. (Japan), Saraya Co., Ltd. (Japan), Lion Corporation (Japan), Kaneka Corporation (Japan), Kao Co., Ltd. (Japan), Mitsubishi India (India), Geocon Products (India), Hindustan Unilever, Ltd. (India), Sasol (South Africa), and others.

The segmentation of the biosurfactant market is carried out according to different criteria. Based on the product type, the global biosurfactant market is divided into (I) glycolipids (sophorolipids and rhamnolipids), (II) lipopeptides and lipoproteins, (III) phospholipids and fatty acids, (IV) polymeric, and (V) others [92]. The sophorolipids segment held the largest share in 2022 [91]. This type of biosurfactant is extensively utilized as ingredients of personal care and cosmetics products. The market of sophorolipids intended only for personal care usage was worth over USD 720 million in 2022 and is expected to grow at an annual rate of 7.5%. Rhamnolipids, another important group of biosurfactants from the glycolipids segment, are widely used in the pharmaceutical, therapeutics, and cosmetic industry, and, due to their unique features, have become progressively favored in a variety of industries. It is estimated that the global rhamnolipid market will be worth over USD 3 billion by 2032 [103].

The global biosurfactant market is segmented by the end-use industry and application into (I) the cleaning industry (industrial cleanser and household detergents), (II) the cosmetics and personal care industry (skin care, hair care, and others products), (III) agriculture (herbicide, insecticide, fungicide, and other agricultural chemicals), (IV) the textile industry, (V) the food processing industry, (VI) the oil and gas industry (oilfield chemicals), (VII) the leather processing industry, (VIII) the pharmaceutical industry, and (IX) others [104]. Household detergents had the largest share of the global biosurfactant market and, based on the research methodology used, it was approximated to have accounted for around 38.9–46.8% in 2022 [91,97].

If the geography is selected as the segmentation criterion, the global biosurfactants market is separated into (I) North America (USA, Canada, and Mexico), (II) Europe (Germany, France, UK, Italy, Netherlands, Spain, Denmark, Belgium, and the rest of Europe), (III) Asia-Pacific (China, Japan, India, South Korea, Australia and New Zealand, Indonesia, Taiwan, Malaysia, and the rest of Asia Pacific), (IV) South America (Brazil, Argentina, Colombia, Chile, and the rest of South America), and (V) the Middle East (Saudi Arabia, UAE, Israel, and the rest of the Middle East) and Africa (South Africa, Nigeria, and the rest of Africa) [104]. Europe is the largest segment of the global biosurfactant market that was valued at USD 1923.06 million in 2022 [97], while Asia Pacific is the fastest growing [105]. According to a study conducted by The Insight Partners, the biosurfactant market in North America accounted for 69.14 kilo tons in 2022 and is expected to grow at an annual rate of 5.5% to reach 105.88 kilo tons by 2030. The development of the biosurfactant market in this region is predominantly driven by the enlarged usage of these compounds in detergents and food processing industries, along with rising environmental concerns, the growing awareness regarding the eco-friendly solution, and the increasing consumer preference toward bio-based cosmetics and personal care products [91].

### 5.2. Dynamics of Biosurfactant Market

An integral part of biosurfactant market analysis is the consideration of the key dynamics, i.e., forces that impact price, demand, and supply, as well as the behaviors of producers and consumers. The factors that are driving the biosurfactant market, prevailing restraints, potential challenges, and future trends are given in Table 4 [90,91,92,97,100,101,103,106].

Although all of the drivers listed in Table 4 notably affect biosurfactant market expansion, the demand for personal care products, detergents, and cleaning agents which are biodegradable and have a low negative environmental impact is the most pronounced due to improving living standards, augmenting population, growing urbanization, increasing consumer awareness regarding health and hygiene, and rising concern about environmental issues. Since the utilization of chemical surfactants is governed by regulations, biosurfactants are increasingly used as an effective green replacement. The incorporation of biosurfactants in the formulations of personal care and hygiene products is in accordance with circular economy principles and consumer preference toward bio-based product usage [91,106]. In addition, given that biosurfactants show upgraded properties compared to chemically derived surfactants, their productive exploitation in oil recovery processes is possible in an environmentally safer way, while biosurfactant consumption in agro-chemical production results in the creation of eco-friendly substances for sustainable agriculture [100,101,107].

The increasing funding for research and development is reflected in the number of published scientific papers in the field. According to the Web of Science [108], there were 8668 publications that included the keywords “Biosurfactants” and/or “Biosurfactant” published between 1996 and 2022. Figure 4 displays the trend of the constantly increasing interest of scientists in the topic of biosurfactants.

The major drawback of biosurfactants is their high price compared to their synthetic counterparts. Chemical surfactants cost around USD 4 per kilogram, while biosurfactants cost around USD 34 per kilogram. Thus, such high costs and, consequently, less biosurfactant commercialization may be the leading factors in restraining market growth. Biosurfactants are expensive due to the high total production cost that is the outcome of a poor yield, as well as the high cost of the raw materials, downstream processing, and storage [91,95,104]. Although using agro-industrial residues and by-products as raw materials might decrease the total production costs, there are no proper cost-effective procedures for the separation and purification of biosurfactants. In addition, the appropriate storage conditions must be met during various stages of the production technology [106]. Moreover, the deficiency of information on toxicity has become the main reason for the limited application of biosurfactants in the food industry. Biosurfactants produced by GRAS microorganisms can potentially be used in the food industry, and medicine as well, but additional investigations are needed [92].

Technological development is a key trend that has become popular in the global biosurfactant market. Biosurfactant-producing companies invest in the development of new cost-effective biosurfactant production technologies that generate novel biosurfactants to fulfill the demands of different end-use industries around the world. Therefore, Evonik Industries AG has launched the sustainable rhamnolipid REWOFERM^®^ RL 100, a novel, fully biodegradable biosurfactant with great cleaning performance produced from local renewable raw materials [102]. Generally, the potential of microorganisms for application in new types of biosurfactant production is unlimited, but a toxicological analysis of the obtained biomolecules should be performed to verify that these compounds are safe for use, especially when the food industry is concerned [92]. Biosurfactant usage in cancer treatment is possible due to their proven anticancer and antimicrobial activity, and the rising number of cancer patients, along with the necessity for safe and nontoxic therapy, can significantly encourage this global biosurfactant market extension. Therefore, this prospective implementation of biosurfactants is currently a primary trend in the global market [106]. Utilizing biosurfactants in the phytoremediation of hydrocarbon-polluted soil occurs as another achievable approach for their application, but the necessary step that precedes its expansion is the determination of the biosurfactants’ toxicity to plants [92].

There are a few challenges associated with biosurfactant production and their presence on the global market (see Table 4). Primarily, it is crucial to develop an economically and environmentally acceptable biotechnological solution for biosurfactant production to sustain the market for these valuable bioproducts and support long-term sustainable development [102].

## 6. Conclusions

In recent decades, sustainable development has gained more and more importance due to the increased awareness of environmental protection and the inevitable depletion of fossil fuels and natural resources, as well as the numerous benefits for society resulting from reduced environmental pollution. Sustainable agriculture and industrial production, and the health of people, animals, and the environment, among other things, largely depend on the adequate management of waste and wastewater. Circular agro-industrial production and bio-economic waste management are of great importance for reducing waste, and its reuse or recycling. In a circular bioeconomy, the transformation of waste creates a balance between industrial processes, economic evolution, and environmental security, while improving the use of resources. Biotechnological processes for the production of biosurfactants based on the application of agro-industrial residues and secondary or waste products are directly involved in the protection of the environment according to the principles of sustainable development, because they use residues/waste created in another technological/biotechnological process as a raw material, i.e., solving their potential problem. Various agro-industrial residues and by-products can be utilized for the production of biosurfactants, which can be divided into four groups: agricultural field residues (stalks and straw), agricultural process residues (corncobs, chaff, and husks), industrial residues (peels, pomace, frying oil, and wastewaters), and industrial by-products (molasses, whey, glycerol, bagasse, and oil cakes).

Biosurfactants have a significant place among bioproducts that have a high commercial and wide use value. These surface-active biomolecules are currently gaining popularity in the international market as a sustainable and green alternative to petroleum-based compounds. Biosurfactants are preferred over chemical surfactants due to their lower toxicity, biocompatibility, biodegradability, better environmental performance, multifunctionality, variety of biological properties, and ability to tolerate temperature and pH fluctuations. Regardless of these favorable characteristics, the commercialization of microbial biosurfactants has not yet been fully realized due to the high production costs. The economic aspects of production, substrates, microorganisms, and product separation and purification procedures are the main factors associated with the commercial application of biosurfactants, and the success of biosurfactant production depends on the possibility of developing more profitable bioprocesses with cheaper substrates, i.e., increasing the product yield while reducing the price of raw materials. Increasing productivity from renewable and waste raw materials, along with the stability and biodegradability of products such as biosurfactants, contribute to improving sustainability and the circular economy.

## Figures and Tables

**Figure 1 foods-13-00711-f001:**
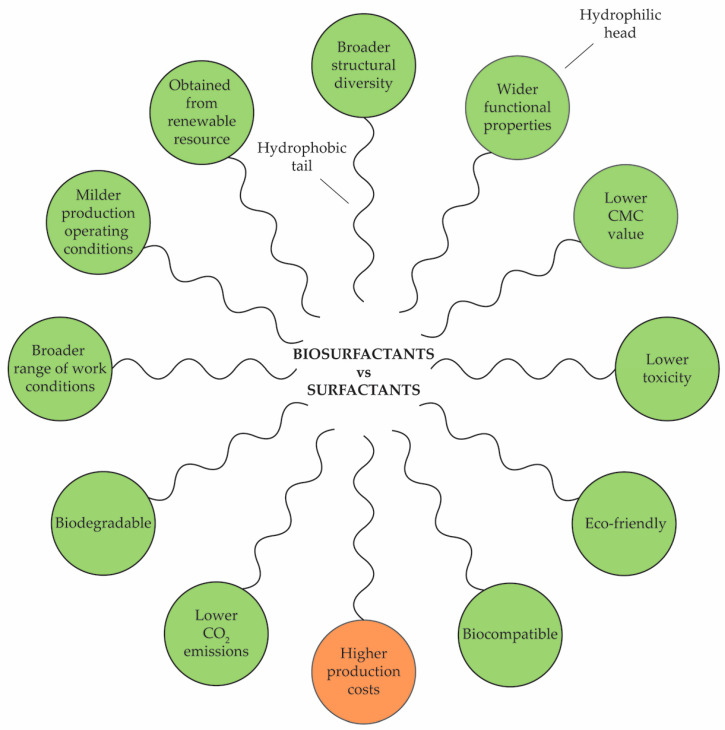
Advantages (green circles) and disadvantages (orange circles) of biosurfactants in comparison to chemically derived surfactants; schematic presentation of a surfactant micelle cluster.

**Figure 2 foods-13-00711-f002:**
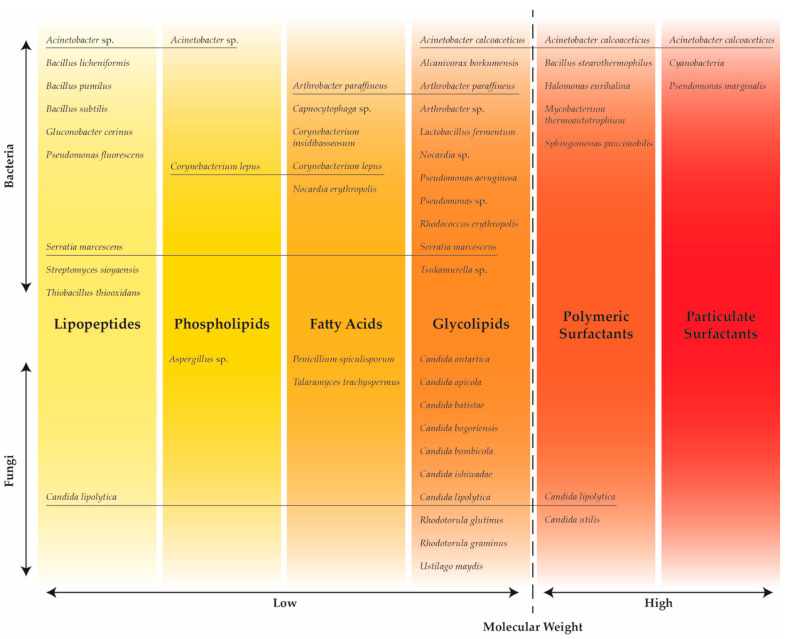
Classification of biosurfactants based on chemical structure and molecular weight, and producing microorganisms.

**Figure 3 foods-13-00711-f003:**
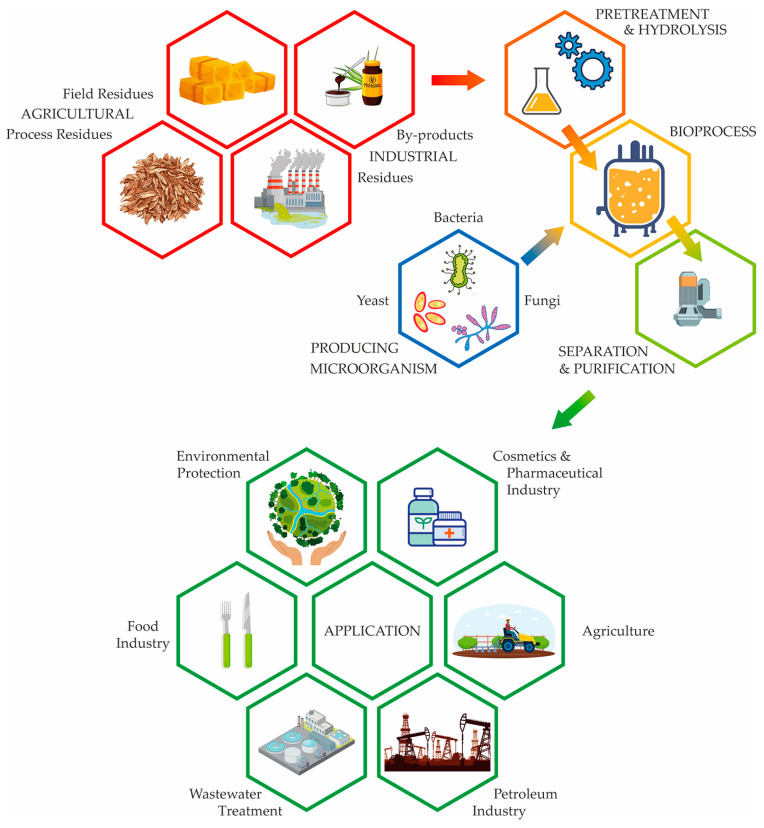
Sustainable production of biosurfactants from agro-industrial residues and by-products.

**Figure 4 foods-13-00711-f004:**
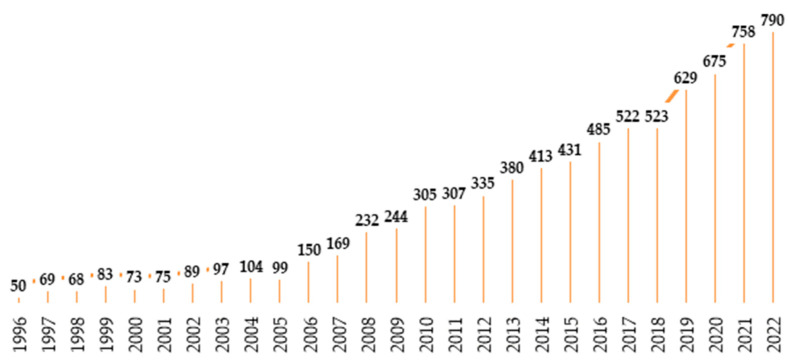
Number of publications related to biosurfactants from 1996–2022. Source: Web of Science [108].

**Table 1 foods-13-00711-t001:** Criteria for consideration when selecting suitable agro-industrial residues and by-products as raw materials in the biotechnological production of biosurfactants.

Type of Agro-Industrial Residue or By-Product	Carbon Source	Biological Availability	Consistency	Seasonal Availability	Alternative Application
Agricultural field residues					
Stalks (corn, sunflower, soya, etc.)	Cellulose Hemicellulose Lignin	Low—requires pretreatment and/or hydrolysis	Solid	Prolonged	Animal feed Fertilizer Heating
Straw (wheat, rice, barley, oat, etc.)	Cellulose Hemicellulose Lignin	Low—requires pretreatment and/or hydrolysis	Solid	Prolonged	Animal bedding Fertilizer Heating
Agricultural process residues					
Corncobs	Cellulose Hemicellulose Lignin	Low—requires pretreatment and/or hydrolysis	Solid	Prolonged	Animal feed Fertilizer Heating
Chaff (wheat, oat, rye, and rice)	Cellulose Hemicellulose Lignin	Low—requires pretreatment and/or hydrolysis	Solid	Prolonged	Animal feed Fertilizer Heating
Husk (coffee, groundnut, etc.)	Cellulose Hemicellulose Lignin	Low—requires pretreatment and/or hydrolysis	Solid	Prolonged	Animal feed Fertilizer Heating
Industrial residues					
Vegetable peels (potato, carrot, etc.)	Starch Pectin Cellulose Hemicellulose	Depends on the present carbon sources	Solid	Very short/All year	Fertilizer Biofuels production
Fruit peels (apple, banana, orange, etc.)	Simple sugars Pectin Cellulose Hemicellulose	Depends on the present carbon sources	Solid	Very short/All year	Fertilizer Biofuels production
Vegetable pomace (tomato, cassava, etc.)	Starch Pectin Cellulose Hemicellulose	Depends on the present carbon sources	Solid	Very short/All year	Food industry
Fruit pomace (grape, apple, etc.)	Simple sugars Pectin Cellulose Hemicellulose	Depends on the present carbon sources	Solid	Very short/All year	Food industry
Frying oil	Triglycerides Diglycerides Monoglycerides Free fatty acids	High—directly consumable	Oily	All year	Biodiesel production
Food processing industry wastewaters	Depends on the processed raw material	Depends on the present carbon sources	Liquid	All year	Biogas production
Industrial by-products					
Molasses (sugar beet, sugarcane, and soy)	Sucrose	High—directly consumable	Syrupy	Prolonged	Bioethanol production Food industry Animal feed
Whey	Lactose	Moderate—consumable by selected strains	Liquid	All year	Protein supplement Food industry Animal feed
Crude glycerol	Glycerol	High—directly consumable	Syrupy	All year	Soap making Substitute for boiler fuel Fuel additive
Bagasse (sugar cane, sweet sorghum, etc.)	Cellulose Hemicellulose Lignin	Low—requires pretreatment and/or hydrolysis	Solid	Very short	Fuel in cogeneration Paper industry
Brewer’s spent grain	Starch Cellulose Hemicellulose Lignin	Depends on the present carbon sources	Solid	All year	Animal feed Bioethanol production
Oil cakes (soybean, sunflower, rapeseed, mahua, olive, etc.)	Starch Cellulose Hemicellulose Lignin Lipids	Depends on the present carbon sources	Solid	Very short/ Prolonged	Animal feed Food industry

**Table 2 foods-13-00711-t002:** Literature review of biosurfactant production using different agro-industrial residues and by-products as sole or partial carbon sources.

Agro-Industrial Residue/By-Product	Pretreatment	Producing Microorganism	Cultivation Conditions	Biosurfactant	Reference
Agricultural field residues					
Sunflower stalks	Crushing, drying, grinding, and alkaline extraction of hemicelluloses	*Bacillus subtilis*	SmF at 30 °C and 120 rpm for 24 h	-	[63]
Rice straw	Drying, milling, and redrying	*Bacillus * *amyloliquefaciens*	SSF at 26.9 °C for 48 h	Surfactin	[64]
Sugarcane straw	Acidic hydrolysis, vacuum concentrating, and detoxification	*Naganishia adellienses*	SmF at 25 °C and 150 rpm for 96 h	-	[65]
Agricultural process residues					
Corncob	Drying, crushing, acidic hydrolysis, and enzymatic saccharification	*Starmerella bombicola*	SmF at 28 °C and 500–1000 rpm for 96–168 h	Sophorolipids	[66]
Corncobs	Crushing, drying, grinding, and alkaline extraction of hemicelluloses	*Bacillus subtilis*	SmF at 30 °C and 120 rpm for 24 h	-	[63]
Industrial residues					
Banana peel	Washing, cutting, drying, and grinding	*Halobacteriaceae archaeon*	SmF at 30 °C and 200 rpm for 48 h	Lipopeptide	[67]
Orange peel	Drying and powdering	*Bacillus licheniformis*	SmF at 30 °C for 120 h	Lipopeptide	[68]
Apple peel extract	-	*Bacillus subtilis*	SmF at 28 °C and 120 rpm for 96 h	Iturin	[69]
Soybean oil waste	-	*Pseudomonas aeruginosa*	SmF at 30 °C and 200 rpm for 240 h	Rhamnolipid	[70]
Waste frying oil	-	*Candida bombicola*	SmF at 30 °C and 0.8 vvm for 14 days	Sophorolipid	[71]
Mango kernel	Drying, grinding, and extraction of oil	*Pseudomonas aeruginosa*	SmF for 120 h	Rhamnolipid	[72]
Industrial by-products					
Sugarcane molasses	Diluting	*Bacillus subtilis*	SmF at 37 °C and 180 rpm for 8 days	Lipopeptide	[73]
Sweet sorghum bagasse	Acidic hydrolysis, and centrifugation for obtaining liquid phase	*Candida (Starmerella) bombicola*	SmF at 25 °C and 120 rpm for 8 days	Sophorolipids	[74]
Mahua oil cake	Drying and grinding	*Serratia rubidaea*	SSF at 30 °C for 7 days	Rhamnolipid	[75]
Sunflower cake	-	*Bacillus subtilis*	SmF at 37 °C and 160 rpm for 168 h	Surfactin	[76]
Rapeseed cake	-	*Bacillus subtilis*	SmF at 37 °C and 160 rpm for 168 h	Surfactin	[76]
Cassava flour wastewater	-	*Serratia marcescens*	SmF at 28 °C and 150 rpm for 72 h	Polymeric surfactant	[77]
Combination of agro-industrial restudies and/or by-products					
Sugarcane bagasse and potato peel	Washing, drying, and grinding	*Pseudomonas azotoformans*	SmF at 35 °C and 180 rpm for 72 h	Rhamnolipid	[78]
Winterization oil cake and sugar beet molasses	-	*Starmerella bombicola*	SSF at 30 °C with aeration rate of 0.30 L kg^−1^ min^−1^ for 10 days	Sophorolipids	[79]
Oil mill wastewater, corn steep liquor, and sugarcane molasses	-	*Pseudomonas aeruginosa*	SmF at 37 °C and 180 rpm for 168 h	Rhamnolipid	[80]
Whey and grape vinasse	Whey: heat deproteinization and centrifugation (to obtain liquid phase); vinasse: centrifugation (to remove solids)	*Lactococcus lactis*	SmF at 37 °C and 100 rpm for 24 h	Glycolipopeptide	[81]

**Table 3 foods-13-00711-t003:** Scope of biosurfactant market.

Market Size * in 2022 (USD Million)	Forecast Period	CAGR ** (%)	Projected Value (USD Million)	Source
4070.00	2023–2027	8.1	6000.00	[90]
4900.00	2022–2027	6.4	6700.00	[92]
1200.00	2023–2028	11.0	2300.00	[93]
1260.00	2022–2028	11.3	2400.00	[94]
4500.00	2022–2028	6.0	6500.00	[95]
2600.00	2023–2028	5.7	3600.00	[96]
3966.42	2022–2029	5.4	6047.39	[97]
5900.00	2023–2029	5.8	8760.00	[98]
811.08	2023–2030	6.2	1311.48	[91]
2051.63	2023–2031	6.1	2925.13	[99]
3700.00	2023–2031	8.2	7500.00	[100]
16.50	2022–2032	3.9	24.30	[101]
1180.00	2022–2032	11.2	3400.00	[102]
8450.00	2023–2032	5.0	14,300.00	[103]

* Market value is defined as the revenues that enterprises gain from the sale of goods and/or services within the specified market and geography through sales, grants, or donations in terms of the currency (in USD). ** CAGR—compound annual growth rate.

**Table 4 foods-13-00711-t004:** Dynamics of biosurfactant market.

Drivers	▪ Growing demand for personal care products. ▪ Growing demand for detergents and cleaning products. ▪ Growing demand for agricultural chemicals. ▪ Rising environmental concern and implementation of circular economy principles. ▪ Growing demand for green solutions and bio-based products. ▪ Regulations governing the exploitation of traditional chemical surfactants. ▪ Advantages associated with the use of biosurfactants. ▪ Increasing use of biosurfactants in the petroleum industry. ▪ Increasing government funding for research and development.
Restraints	▪ High total cost of biosurfactant production. ▪ High price of biosurfactants compared to synthetic surfactants. ▪ Less commercialization of biosurfactants. ▪ Deficiency of information on biosurfactant toxicity.
Trends	▪ Development of cost-effective biosurfactant production technology. ▪ Development of novel biosurfactants to meet the user’s technical demands. ▪ Potential use of biosurfactants in cancer treatment. ▪ Potential use of biosurfactants in the phytoremediation of hydrocarbon-polluted soil.
Challenges	▪ Lack of sustainable biosurfactant production technology. ▪ Safety of workers involved in the biosurfactant production process. ▪ Limitations of biosurfactant structure modification.

## Data Availability

No new data were created or analyzed in this study. Data sharing is not applicable to this article.

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
