# Peer review of "Biotechnological Utilization of Agro-Industrial Residues and By-Products—Sustainable Production of Biosurfactants"

_foods, 2024, doi:10.3390/foods13050711_

Round 1
Reviewer 1 Report
Comments and Suggestions for Authors
It is important information in the biotechnology area and in applications, which is why I consider it suitable for this journal, after the authors make the comments and suggestions that I mention below. My decision is minor revisión
Comments to authors
1. In the introduction section, further justify the importance of using waste and byproducts by adding the following paragraph:
"Food industries generate a large amount of agri-food waste at the level of agricultural crops, as well as during processing, with around 1.6 billion tons of primary product equivalents, and the total food waste for the edible part of this amounts to 1.3 billion tons (Rodríguez-Félix et al., 2022). The advantage of using materials from the agri-food industry (waste or by-products) as raw material decreases the cost of production when carried on an industrial scale. This is due to the fact that the waste does not have a potential use, production costs can be lowered.
- (2022 ). Physicochemical, structural, mechanical and antioxidant properties of zein films incorporated with non-ultrafiltered and ultrafiltered betalains extract from the beetroot (Beta vulgaris) bagasse with potential application as active food packaging. Journal of Food Engineering, 334, 111153.
2. Make a table or figure that describes the differences (advantages and disadvantages) between surfactants and biosurfactants, this with the purpose of better understanding this chapter of the review article.
3. Include a figure at the beginning as a graphic abstract that describes the review manuscript.
4. Include in the legend of figure 3 from which statistical database the information on biosurfactants was collected by year.
5. In the conclusion part, write about the waste and byproducts for the production of biosurfactants
Author Response
We thank the reviewer for all comments and suggestions made during the review of this manuscript. After carefully reading the comments, all suggestions were accepted, and we have made the necessary changes in our manuscript. Detailed responses to each issue raised by the reviewer are provided below.
Point 1: In the introduction section, further justify the importance of using waste and byproducts by adding the following paragraph:
"Food industries generate a large amount of agri-food waste at the level of agricultural crops, as well as during processing, with around 1.6 billion tons of primary product equivalents, and the total food waste for the edible part of this amounts to 1.3 billion tons (Rodríguez-Félix et al., 2022). The advantage of using materials from the agri-food industry (waste or by-products) as raw material decreases the cost of production when carried on an industrial scale. This is due to the fact that the waste does not have a potential use, production costs can be lowered.
- (2022 ). Physicochemical, structural, mechanical and antioxidant properties of zein films incorporated with non-ultrafiltered and ultrafiltered betalains extract from the beetroot (Beta vulgaris) bagasse with potential application as active food packaging. Journal of Food Engineering, 334, 111153.
Response 1: We thank the reviewer for this suggestion. We added the proposed paragraph to the Introduction section in order to further justify the importance of using waste and byproducts. The only modification we made was to put the reference at the end of the paragraph and relatively change the last two sentences to be written in line with the rest of the paragraph:
“Food industries generate a large amount of agri-food waste at the level of agricultural crops, as well as during processing, with around 1.6 billion tons of primary product equivalents, and the total food waste for the edible part of this amounts to 1.3 billion tons. When produced on an industrial scale, the benefit of utilizing waste or by-products from the agri-food industry as raw materials is a reduction in manufacturing costs, due to the fact that the waste does not have a potential use [7].”
Additionally, the mentioned reference was added to the list of references, under the number 7, as a result of which the numbering of the references was shifted by one:
- Rodríguez-Félix, F.; Corte-Tarazón, J.A.; Rochín-Wong, S.; Fernández-Quiroz, J.D.; Garzón-García, A.M.; Santos-Sauceda, I.; Plascencia-Martínez, D.F.; Chan-Chan, L.H.; Vásquez-López, C.; Barreras-Urbina, C.G.; Olguin-Moreno, A.; Tapia-Hernández, J.A. Physicochemical, structural, mechanical and antioxidant properties of zein films incorporated with no-ultrafiltered and ultrafiltered betalains extract from the beetroot (Beta vulgaris) bagasse with potential application as active food packaging, J. Food Eng. 2022, 334, 111153. https://doi.org/10.1016/j.jfoodeng.2022.111153
Point 2: Make a table or figure that describes the differences (advantages and disadvantages) between surfactants and biosurfactants, this with the purpose of better understanding this chapter of the review article.
Response 2: We thank the reviewer for this comment and we added the new figure with the follwing caption: “Figure 1. Advantages (green circles) and disadvantages (orange circles) of biosurfactants in comparison to chemically derived surfactants; schematic presentation of a surfactant micelle cluster”, wich summarizes the differences (advantages and disadvantages) between surfactants and biosurfactants from Section 2.
Point 3: Include a figure at the beginning as a graphic abstract that describes the review manuscript.
Response 3: The graphic abstract (similar to Figure 4) is sent as a separate document during the submission of the manuscript. If necessary, we can include it in the manuscript document itself, but now when sending the revised manuscript, we will upload it again to the section provided for the graphical abstract.
Point 4: Include in the legend of figure 3 from which statistical database the information on biosurfactants was collected by year.
Response 4: We thank the reviewer for this comment and we added to Figure 3 (now Figure 4) that the data were obtained from the Web of Science website, with the reference: https://www.webofscience.com/wos/woscc/summary/abf01f1d-764e-42e5-a709-a16e8002c1bb-b780253b/relevance/1
Point 5: In the conclusion part, write about the waste and byproducts for the production of biosurfactants
Response 5: We thank the reviewer for this comment and we have added the corresponding paragraph about waste and by-products for the production of biosurfactants in the Conclusion part:
“Various agro-industrial residues and by-products can be utilized for the production of biosurfactants, which can be divided in four groups: agricultural field residues (stalks, straw), agricultural process residues (corncobs, chaff, husk), industrial residues (peels, pomace, frying oil, wastewaters), and industrial by-products (molasses, whey, glycerol, bagasse, oil cakes).”
Reviewer 2 Report
Comments and Suggestions for Authors
1、Compared with the reviews of Mohanty et. al and Santos et. al, what is/are the main innovation(s) of your work?
2、L123-L139: the content of this paragraph is tedious and needs to be more concise.
3、The authors reviewed the production of biosurfactants by a wide variety of distinct microorganisms, however, the corresponding processing parameters and production steps need to be introduced.
4、In Table 2, there are no types of biosurfactants produced from Sunflower stalks, Sugarcane straw and Corncobs as raw materials, please explain this.
5、I think that adding different subtitles for each section will make the structure of the paper clearer.
6、Please check and standardize the format of references.
Comments on the Quality of English LanguageThe article writing is good.
Author Response
We thank the reviewer for all comments and suggestions made during the review of this manuscript. After carefully reading the comments, all suggestions were accepted, and we have made the necessary changes in our manuscript. Detailed responses to each issue raised by the reviewer are provided below.
Point 1: Compared with the reviews of Mohanty et. al and Santos et. al, what is/are the main innovation(s) of your work?
Response 1: Compared to other review papers dealing with the topic of production of biosurfactants from agro-industrial wastes, residues and by-products (among which are review papers of the mentioned authors), the innovation of this paper is reflected in the fact that a detailed classification is given as well as the selection criteria of various agro-industrial residues and by-products that can be used for the sustainable production of biosurfactants, as well as examples of the production of biosurfactants using various agro-industrial residues and by-products. Additionally, based on the available biosurfactants market analysis datasets and research studies, the current situation in science and industry and future perspectives of microbial biosurfactant production have been discussed.
Point 2: L123-L139: the content of this paragraph is tedious and needs to be more concise.
Response 2: We agree with the reviewer's comment regarding the above paragraph, which was clarified for the above reason:
“The critical micelle concentration (CMC) refers to the minimum concentration of a surfactant in a bulk phase at which micelles, which are aggregation of surfactant molecules, begin to form. In these micelles the hydrophilic heads of the surfactant molecules are distributed along the perimeter of the sphere, while the hydrophobic tails are pointed toward its core (schematic presentation in Figure 1). In many processes the CMC specifies the limiting concentration for meaningful use. When the formation of micelles is desirable, e.g. when cleaning, the CMC is a measure of the efficiency of a surfactant. Surfactants with lower CMC values are more effective. Biosurfactants have 40% lower CMC than synthetic surfactants [38]. The ratio of the hydrophilic portion of the molecule to the hydrophobic portion, known as the Hydrophilic-Lipophilic Balance (HLB), affects the efficacy of surfactants' wetting, anti-foaming, and emulsifier properties [46].”
Point 3: The authors reviewed the production of biosurfactants by a wide variety of distinct microorganisms, however, the corresponding processing parameters and production steps need to be introduced.
Response 3: We thank the reviewer for this suggestion and in order to respond to this comment we have added columns containing Pretreatment of raw material, and Cultivation conditions of biosurfactant production in Table 2.
Point 4: In Table 2, there are no types of biosurfactants produced from Sunflower stalks, Sugarcane straw and Corncobs as raw materials, please explain this.
Response 4: The reason for which the biosurfactant in question is not specified in the above-mentioned two papers is because its detailed classification was not done. Namely, appropriate analyzes were performed that confirm that it is a biosurfactant, but its composition is not analyzed and determined, and therefore it is not classified.
Point 5: I think that adding different subtitles for each section will make the structure of the paper clearer.
Response 5: We agree with the reviewer suggestion and we added several subtitles in order to clarify the structure of the paper. Titles of the chapters and added subchapters are as follows:
- Surfactants vs. biosurfactants
2.1. Adventages and disadventages
2.2. Classification of biosurfactants
- Agro-industrial residues and by-products for biosurfactants production
3.1. Clasification of agro-industrial residues and by-products
3.2. Agro-industrial residues and by-products selection criteria
- Current State and Future Perspectives of Biosurfactants Production
5.1. Biosurfactants market
5.2. Dynamics of biosurfactants market
Point 6: Please check and standardize the format of references.
Response 6: We thank the reviewer for this comment, the references have been written in accordance with the journals instructions and the observed formatting errors have been corrected.
Round 2
Reviewer 2 Report
Comments and Suggestions for Authors
I agree to publish this paper.